# Mathematical Modeling of Non-Small-Cell Lung Cancer Biology through the Experimental Data on Cell Composition and Growth of Patient-Derived Organoids

**DOI:** 10.3390/life13112228

**Published:** 2023-11-20

**Authors:** Rushan Sulimanov, Konstantin Koshelev, Vladimir Makarov, Alexandre Mezentsev, Mikhail Durymanov, Lilian Ismail, Komal Zahid, Yegor Rumyantsev, Ilya Laskov

**Affiliations:** 1Medical Informatics Laboratory, Yaroslav-the-Wise Novgorod State University, 173003 Veliky Novgorod, Russia; sulimanov@mail.ru (R.S.); k.koshelev@ispras.ru (K.K.); vladimir.makarov@novsu.ru (V.M.); mesentsev@yahoo.com (A.M.); durymanov.mo@mipt.ru (M.D.); laskovi.workmail@gmail.com (I.L.); 2Ivannikov Institute for System Programming of the Russian Academy of Science, 109004 Moscow, Russia; 3School of Biological and Medical Physics, Moscow Institute of Physics and Technology, 141701 Dolgoprudny, Russia; ismail.l@phystech.edu (L.I.); komalzahid@phystech.edu (K.Z.)

**Keywords:** mathematical modeling, non-small-cell lung cancer, flow cytometry, cell composition, tumor-associated macrophages, cytotoxic T-lymphocytes, adenocarcinoma, cancer cells, cancer-associated fibroblasts

## Abstract

Mathematical models of non-small-cell lung cancer are powerful tools that use clinical and experimental data to describe various aspects of tumorigenesis. The developed algorithms capture phenotypic changes in the tumor and predict changes in tumor behavior, drug resistance, and clinical outcomes of anti-cancer therapy. The aim of this study was to propose a mathematical model that predicts the changes in the cellular composition of patient-derived tumor organoids over time with a perspective of translation of these results to the parental tumor, and therefore to possible clinical course and outcomes for the patient. Using the data on specific biomarkers of cancer cells (PD-L1), tumor-associated macrophages (CD206), natural killer cells (CD8), and fibroblasts (αSMA) as input, we proposed a model that accurately predicts the cellular composition of patient-derived tumor organoids at a desired time point. Combining the obtained results with “omics” approaches will improve our understanding of the nature of non-small-cell lung cancer. Moreover, their implementation into clinical practice will facilitate a decision-making process on treatment strategy and develop a new personalized approach in anti-cancer therapy.

## 1. Introduction

Non-small-cell lung cancer (NSCLC) is one of the deadliest forms of cancer worldwide. Lung cancer organoids are three-dimensional cell aggregates grown from tumor cells in vitro. Unlike traditional monolayered cell cultures, patient-derived tumor organoids (PDTOs) preserve cellular architecture, mutations, and growth of their parental tumor [1,2]. In this respect, exploring the properties of patient-derived organoids represents an attractive option to explore tumor organization, evolution, and complex behavior in controlled conditions.

Mathematical models are often used to evaluate drug responses in infectious and oncologic disorders to predict the outcome of proposed therapy [3,4]. They are also used to assess the efficacy of vaccines. In this regard, nearly any signaling or metabolic pathway is describable as a combination of mathematical equations in the system [5,6,7]. Previously, several authors have proposed mathematical models of tumor growth (rev. in [8]). In such mathematical models, an ordinary differential equation (ODE) is an equation for a function of an independent variable that involves several derivatives characterizing its behavior. The systems of differential equations make it possible to address more complex problems. For instance, it would be possible to assess tumor size using the data on its cellular composition. In perspective, advanced mathematical models will define a time window most suitable for the surgery.

To date, several known mathematical models have been used to predict the malignancy of lung cancer. Some of these models use imaging data and the results of clinical tests (rev. in [9]), while others use the patients’ genetics, such as mutations and specific biomarkers. In addition, several mathematical models of NSCLC address the changes in cell composition of tumors. However, some describe changes in selected subpopulations of cells; others consider changes in the proliferation and apoptosis of tumor cells [10] and penetrability of tumors to cells and drugs [11], not counting on their cellular diversity. For instance, Eftimie R. and Barelle C. [12] derived a mathematical model considering the interactions between phenotypically diverse macrophages. At the same time, this is not the only cell type of tumor-associated cells. In turn, the ODE (ordinary differential equation) model proposed by Lourenço E. Jr. et al. described the interaction of macrophages and CD8-positive cytotoxic cells in the microenvironment of NSCLC tumors [13]. Molina-Peña R. et al. developed a mathematical model that links tumor growth and relapse to the dynamics and interaction of cancer stem cells and progenitor cells [14]. As believed, efficient mathematical models with good prediction performance will improve the detection of lung cancer at early stages due to the facilitation of its diagnosis.

Unlike the other researchers, we consider the changes in four main subpopulations of cells represented in NSCLC tumors: cytotoxic CD8-positive cells, tumor-associated macrophages (M2/ CD206-positive cells), PD-L1-positive cancer cells, and α-SMA-positive tumor-associated fibroblasts. In our study, we apply a computational approach to assess the parameters of a mathematical model simulating the time-dependent counts of cellular subpopulations in patient-derived organoids. The proposed mathematical model originates from the models of lung cancer progression previously developed by Geng et al. [15]. Using this approach, we systematically explored changes in the cell composition of PDTOs. Our aim was to develop a mathematical model for the prediction of changes in the cellular composition of organoids with a perspective of translating these results into the respective parental tumor which potentially can be used as a part of patient condition assessment and NSCLC prognosis.

We implemented our mathematical model as an original software application. However, it still needs improvement. For instance, more experiments are necessary to increase the preciseness and accuracy of the predictions made, as well as readjust previously made assessments of its parameters. We would also consider the inclusion of other tumor-associated cells, like neutrophils, in the developed mathematical model.

## 2. Materials and Methods

### 2.1. Patients

The material was obtained from 16 patients from the first oncological hospital (Moscow, Russia) and Novgorod Regional hospital (Veliky Novgorod, Russia) after surgery. The mean age was 65 years (SD 4.1); 13 of the patients were men (81%) and 3 were women (19%). All the patients had undergone thoracotomic lobectomy due to non-small-cell lung cancer (NSCLC). A mathematical model was developed based on data obtained from patient-derived organoid culture of NSCLC.

### 2.2. Generation of PDTOs

Tumor samples were placed into basal DMEM/F12 (PanEco, Moscow, Russia) culture medium supplemented with 1× penicillin–streptomycin (PanEco) and delivered to the lab on ice immediately after the surgery. In the lab, the tissue was finely chopped into small pieces using surgical scissors and then immersed in 10 mL of DMEM/F12 growth medium. This medium was supplemented with 1% penicillin–streptomycin (PenStrep) and 1 mg mL^−1^ of collagenase I (Life Technologies, Waltham, MA, USA). Afterward, the minced tissue was allowed to incubate at 37 °C for 1.5 h with gentle and slow agitation.

Following the incubation, the digested tissue suspension was filtered through a 70 μm cell strainer (Corning, NY, USA). The resulting cell suspension was then subjected to centrifugation at 1500 rpm for 5 min at a temperature of 18–20 °C. The pellet obtained after centrifugation was washed with HBSS and subsequently resuspended in 6 mL of fresh growth medium suitable for organoid culture. This organoid culture medium consisted of DMEM/F12 supplemented with 20 ng mL^−1^ of basic fibroblast growth factor (bFGF, 10014-HNAE, Sino Biological, Beijing, China), 50 ng mL^−1^ of human epidermal growth factor (EGF, ab55566, Abcam, Cambridge, UK), N2 supplement (PanEco, Moscow, Russia), NeuroMax (PanEco, Moscow, Russia), 10 mM Glutamax (Gibco), 1 mM N-acetyl cysteine (Merck, Burlington, MA, USA), 10 mM nicotinamide (Sigma), 10 μM Y27631 (ab120129, Abcam), 15 μM HEPES (Merck), and 1% PenStrep. The total cell count was determined using a hemocytometer, and a portion of the collected cells was preserved by freezing for subsequent flow cytometry analysis.

To generate free-floating NSCLC organoids, 96-well plates were initially coated with a 1% *w*/*v* agarose solution in Milli-Q water, applying 50 μL to each well. This agarose coating was allowed to solidify at room temperature for approximately 20–30 min. Subsequently, a collagen-based gel solution containing resuspended tumor cells was prepared using the SANATO 3D culture gel kit (#FTBM0051, Phystech Biomed, Dolgoprudny, Russia) and 1% (*v*/*v*) SANATO reagent (#FTBM0050, Phystech Biomed, Russia) following the manufacturer’s protocol. Next, 25 μL gel domes, each containing 50,000 cells, were dispensed into individual wells, and the gel was allowed to solidify at 37 °C for 20 min. After the gel had set, 100 μL of organoid growth medium was added to each well, and the plates were placed in an incubator. The incubation was carried out at 37 °C in a humidified atmosphere with 5% CO_2_. The growth medium was replaced every other day throughout the entire experiment. Over a period of 14 days, the organoids were subject to daily examination, during which their size and morphology were observed using bright-field microscopy with an AxioVert.A1 microscope (Zeiss, Oberkochen, Germany).

### 2.3. Flow Cytometry

Harvested organoids were centrifuged (1500 rpm, 5 min, r.t.). The pellet was resuspended in basal DMEM/F12 cell culture medium (1 mL/plate). Then, an equal volume of 0.1% collagenase I was added and the samples were incubated for 90 min at 37 °C with shacking. After the incubation, the cells were washed and resuspended in Versene (0.48 mM EDTA, pH 7.4). The resuspended cells were fixed in 4% formaldehyde for 15 min at room temperature and washed twice in excess of 1× PBS. After washing, the cells were resuspended in 0.1% TRITON X-100 prepared in 1× PBS and incubated for 15 min at room temperature. Then, the cells were incubated with primary antibodies for 1 h in the dark and on ice. The following primary antibodies were used in this study: rabbit Alexa Fluor^®^ 647 monoclonal anti-human antibodies directed to mannose receptor/CD206 (ab195192, Abcam), mouse monoclonal anti-human antibodies directed to CD8α (ab33786, Abcam), rabbit recombinant anti-human antibodies directed to PD-L1 (ab205921, Abcam), and rabbit recombinant anti-human antibody directed to α-smooth muscle actin/αSMA (ab124964, Abcam). After the incubation, the cells were washed twice in ice-cold Versene and resuspended in 1× PBS. Then, the samples were incubated with secondary antibodies for 30 min in the dark and on ice. The following secondary antibodies were used: goat anti-rabbit IgG H&L conjugated with Alexa Fluor^®^ 647, preadsorbed (ab150083, Abcam), and goat anti-mouse IgG H&L conjugated with Alexa Fluor^®^ 647 (ab150115, Abcam). After the incubation, the cells were washed with Versene and subjected to flow cytometry.

### 2.4. Mathematical Approaches Used in This Study

The obtained results were analyzed on a PC using Python 3.7. The libraries numpy 1.24.2 and maptplotlib 3.7.1 were used for data processing. Scipy 1.10.1 was used to solve the differential equation system.

## 3. Results

### 3.1. General Work Flow

In this study, we present a diagram (Figure 1) to visualize the interactions of cells within NSCLC tumors and generated the tumor-specific microenvironment based on the previously published data on their cell composition [16]. We converted the diagram into the system of the differential Equation (1) to describe the changes in subpopulations of cells in time. We also performed flow cytometry analysis of PDTOs to identify four main subpopulations of cells expressing the specific biomarkers (Table 1). Then, we solved the system of differential equations using the assessments of several parameters previously reported by others (Table 2).

### 3.2. Mathematical Model

The mechanistic model describing the interaction of cells in NSCLC tumors is defined by the system of the differential Equation (1): (1)$$\left\{\begin{array}{c} \frac{dN}{dt}=\gamma \left(1-\frac{N}{K}\right)N+Nq_{1}M2+Nq_{2}CAF-kT_{c}N,\\ \frac{dM2}{dt}=M2q_{3}N+M2q_{4}CAF-\delta_{M2}M2,\\ \frac{dCAF}{dt}=CAFq_{5}N+CAFq_{6}M2-\delta_{CAF}CAF,\\ \frac{dT_{c}}{dt}=T_{c}q_{7}N-T_{c}q_{8}M2-T_{c}q_{9}CAF-\delta_{T_{c}}T_{c} \end{array}\right.$$
where *t*—time in days, *N*—number of cancer cells, *M*2—number of *M*2-polarized (tumor-associated) macrophages, *CAF*—number of cancer-associated fibroblasts, and *T_c_*—number of cytotoxic T cells. The values of *γ*, *K*, *q*_1_, *k*, *q*_3_, *δ_M_*_2_, and *δ_Tc_* were previously assessed by others (see Table 2). The remaining parameters, namely *q*_2_, *q*_4_, *q*_5_, *q*_6_, *δ_CAF_*, *q*_7_, *q*_8_, and *q*_9_, were assessed by us using the quantity of cells measured in PDTOs on days 7, 14, and 21 via flow cytometry for PDTOs developed from various tumors. Performing the calculations, we assumed that all assessed parameters had to be above zero. The results of calculations are represented in Table 3.

### 3.3. Solving the System of Differential Equations

The system of differential Equation (1) can be written in vector form $\frac{dZ}{dt}=F(Z)$, where
$$z=\left(\begin{array}{c} N\\ M2\\ CAF\\ T_{c} \end{array}\right)$$

In the time interval between *i* and *i* + 1, the system of differential equations can be approximated by the system of difference equations in form (2) using the experimental data obtained for individual tumors.
(2)$$\frac{Z^{i+1}-Z^{i}}{t^{i+1}-t^{i}}=F(Z^{i+\frac{1}{2}})$$

Vector $Z^{i+\frac{1}{2}}$ was calculated using interpolation Formulas (3) and (4) for non-negative and strictly positive elements, respectively.
(3)$$Z_{j}^{i+\frac{1}{2}}=\frac{Z_{j}^{i}+Z_{j}^{i+1}}{2}$$
(4)$$Z_{j}^{i+\frac{1}{2}}=\frac{2\cdot Z_{j}^{i}\cdot Z_{j}^{i+1}}{Z_{j}^{i}+Z_{j}^{i+1}}$$

*j* is the number of the element in the vector *Z*.

Since all parameters that we needed to estimate were introduced in Equation (2) linearly, this system was written as a system of linear Equation (5)
*AQ* = *B*(5)

In this case, *A* and vector *B* were calculated from the terms of Equation (2). Elements of the vector *Q* were the parameters that we needed to estimate. The number of columns of the matrix *A* and dimensions of the vector *Q* were equal to 8, i.e., the number of parameters that we needed to estimate. The number of rows in the matrix *A* and dimensions of the vector *B* depended on how many measurements were made and should not be less than 8. Since the system of Equation (5) was overdetermined, it was solved using the least-squares method:*Q* = (*A^T^* × *A*)^−1^ × *A^TB^*(6)

The results of calculations are presented in Table 3.

To implement the above methodology, we developed a program in Python. The program accepts input data (the results of flow cytometry experiments performed in two time points at least) in Microsoft Excel format and predicts changes in cell composition of PDTOs. To avoid negative values for the parameters listed in Table 3, we adjusted the published values of known parameters as indicated in the last column of Table 2.

Using the proposed mathematical model, describing the interactions in four subpopulations of tumor cells, we explored how the cellular composition of PDTOs could change from day 7 to day 14 of the experiment (Figure 2). The results demonstrated that the predicted changes in subpopulations of cells, except the ratio of PDL1-positive cells in the sample donated by patient 8, were describable by functions without local extrema between the desired time points (Figure 2). In the last case (Figure 2a), the ratio of PDL1-positive cells kept increasing from day 7 and reached the maximum at day 11 of the experiment. In the sample donated by patient 9, the changes in the subpopulation of CD206-positive cells did not exceed 2%, whereas the ratio of αSMA-positive cells changed by a fraction of a percent (Figure 2b). In samples with a prevalence of PDL1- and CD8-positive cells (patients 13 and 14, the predicted changes in subpopulations of other cells (αSMA, CD206, and CD8 in the former and αSMA, CD206, and PDL1 in the latter cases, respectively), did not exceed a fraction of a percent (Figure 2c,d). Contrarily, the ratios of PDL1- and CD8-positive cells in samples 13 and 14 progressively increased by factors of 1.5 and 1.4, respectively.

## 4. Discussion

In this paper, we analyzed changes in four subpopulations of cells in PDTOs originating from NSCLC tumors. PD-L1-positive cells represented a prominent cancer phenotype capable of suppressing the adaptive arm of immune systems. The cells expressing αSMA were typically of mesenchymal origin. The authors of experimental studies often considered αSMA as a specific biomarker of CAFs. NSCLC tumor cells positive for CD206 and CD8 represented tumor-associated macrophages and cytotoxic T-cells.

Similar to other cell-specific biomarkers, the expression of PD-L1 also occurs in healthy cells, such as immune and epithelial cells, particularly under inflammatory conditions, such as autoimmune responses, chronic infection, and sepsis [20]. The binding of PD-L1 to the inhibitory checkpoint molecule PD-1 located on the surface of activated immune cells activates an inhibitory signal that helps the targeted cells bypass surveillance by immune cells. In this regard, the expression of PD-L1 by cancer cells lets them evade the host immune system. Recruiting stromal and immune cells allows cancer cells to bundle up an immunosuppressive microenvironment. This tumor-specific microenvironment is suitable for the proliferation of cancer cells. It also facilitates the development of drug resistance (Figure 1).

The expression of αSMA mainly occurs in connective tissues in mesenchymal cells, such as vascular endothelial cells and smooth muscle cells. In fibroblasts, αSMA becomes induced upon their activation following an injury. In the lung, the presence of αSMA-positive fibroblasts is evident in fibrotic tissues and tumors [21]. As a part of a tumor, the activated fibroblasts establish direct contact with the cancer cells. Moreover, cancer cells influence their gene expression via soluble factors promoting the acidification of the surrounding milieu and establishing the hypoxic conditions in the core of the tumor. Participating in the secretion and deposition of the extracellular matrix [22] by tumors, CAFs provide them with structural support and improve their resistance to anti-cancer drugs (Figure 1).

The mannose receptor CD206 is a surface receptor of M2 macrophages. Unlike CD86-positive (M1) macrophages participating in the development of inflammatory response, M2 macrophages suppress inflammation and contribute to wound healing [23]. The polarization of macrophages to either the M1 or M2 phenotype is reversible. For instance, recruiting M1 macrophages by tumor cells induces their repolarization to the M2 phenotype [24]. After repolarization, M2 macrophages become tumor-associated macrophages (TAM). Sustained in the tumor microenvironment, TAMs promote tumor growth and cancer progression (Figure 1) by producing various growth factors and cytokines, such as IL6, FGF, and VEGFA. These cytokines stimulate angiogenesis, tumor cell proliferation, and invasion. Expressing immunosuppressive cytokines (e.g., IL10 and TGFβ), TAMs suppress the activation of T cells. TAMs also contribute to the production of extracellular matrix. In turn, covering the tumor with a layer of extracellular protects tumor cells from apoptosis and improves their resistance to anti-cancer therapies.

The CD8 protein is a specific receptor of cytotoxic T lymphocytes. This receptor participates in their interaction with antigen-presenting cells. In the body, cytotoxic CD8-positive T cells kill pathogens and eliminate cancer cells [25]. However, some CD8-positive T cells acquire unresponsiveness to cancer cells due to prolonged exposure to TCR signaling via the cognate antigen. Chronic exposure to the antigen in the tumor microenvironment induces the genes of inhibitory receptors, such as PD-1. Although these cells remain in the tumor microenvironment, they stop expressing the proinflammatory cytokines (IFNγ, TNF, and IL2) and rarely proliferate [26]. The exhausted CD8-positive cells contribute to tumor survival by driving tumor cells to impair immune attack and recruiting other cells to reprogram the immune milieu. Interaction with CAFs also suppresses T-lymphocyte activity [27] by inhibiting the differentiation of CD8-positive T cells and disabling their tumor reactivity to benefit the cancer cells (Figure 1).

The mathematic model reported in this study allowed us to estimate the scale of changes in the four most abundant subpopulations of cells previously discovered in NSCLC tumors (Table 3) and create a computer program to predict changes in cellular composition of PDTOs using the results of flow cytometry experiments performed at two data points (Figure 2). We found that the prevalence of some cells in PDTOs over the others suppresses their growth (Figure 2c,d). Cytotoxic CD8-positive cells may interfere with PDL1-positive cells, mostly cancer cells (Figure 2a,b). In Figure 2a, the ratio of PDL1-positive cells in the sample kept declining starting from day 11 of the experiment. On the other hand, the fraction of CD8-positive cells rapidly increased. The opposite scenario occurred in PDTOs donated by patient 9 (Figure 2b), where cytotoxic CD8-positive cells became nearly undetectable on day 14. Contrarily, the ratio of PDL1-positive cells stabilized at 3% after a steady decline. In turn, the deviations between some predicted values and experimental data indicated that the developed computer program needs improvement.

According to the previously published data, some antigens considered in our study, such as α-SMA and PDL1 (Table 2), are not strictly limited to CAFs and cancer cells, respectively. In this regard, some cancer cells do not express PD-L1. However, expressing PD-L1 helps cancer cells to avoid immune surveillance [28,29]. Moreover, a certain percentage of M2 also expresses PD-L1 [30]. Therefore, the number of PD-L1-expressing macrophages shall be subtracted from the total number of PD-L1-positive cells. In turn, some cancer cells express α-SMA because α-SMA is one of the known biomarkers of epithelial–mesenchymal transition. However, the number of non-fibroblast α-SMA-positive cells in our experimental conditions was negligible by measuring the cellular composition in the experimental block. In addition to the above, a systematic literature review was conducted throughout the model study pathway to more accurately reproduce intercellular interactions. Quantification of double-labeled cells and taking them into account would improve the accuracy of the made predictions.

In turn, the presented mathematic model also has several limitations. First, we performed this study on a limited number of patients. A higher sample size would improve the accuracy of our predictions. Second, our prediction power is time-limited since we used the experimental data obtained at three different time points—on days 7, 14, and 21 of the experiment. Respectively, moving out of this time interval will lower the accuracy of our assessments. Third, although organoids are similar to their parental tumors in many aspects [31], their prolonged culturing in vitro would result in different clinical phenotypes due to the isolation of organoids from the host immune and endocrine system. The organoids do not experience pressure from the approaching immune cells. The growth factors that the organoids receive with culture medium are not necessarily the same and in the same concentrations as the ones delivered to the tumor in vivo. [32]. Fourth, the ability of organoids to recruit new cells is limited to the cells already present in the well at plating. Fifth, the model does not consider changes in minor subpopulations of cells.

## 5. Conclusions

The proposed mathematical model allowed us to range previously unknown parameters characterizing the intercellular interactions in PDTOs obtained from NSCLC tumors. The following implementation of this model into the computer application lets us predict the changes in the four most abundant subpopulations of NSCLC tumor cells: PD-L1-positive cancer cells, α-SMA-positive CAFs, CD206-positive TAM, and CD8-positive cytotoxic T-lymphocytes. Although the proposed mathematic model has several limitations and the developed software still needs improvement, in perspective, they would fit into clinical practice to estimate the probability of tumor relapse and survivability of the patients. At the time, we are planning a follow-up study on a new cohort of patients to increase the accuracy of computer applications. We anticipate that the improved version of the computer application would be able to contribute to a solid foundation for developing new NSCLC models, confirm the results of diagnostic tests, and make predictions on the cellular composition of tumors using the available clinical data of individual patients, with no need for additional statistical data.

## Figures and Tables

**Figure 1 life-13-02228-f001:**
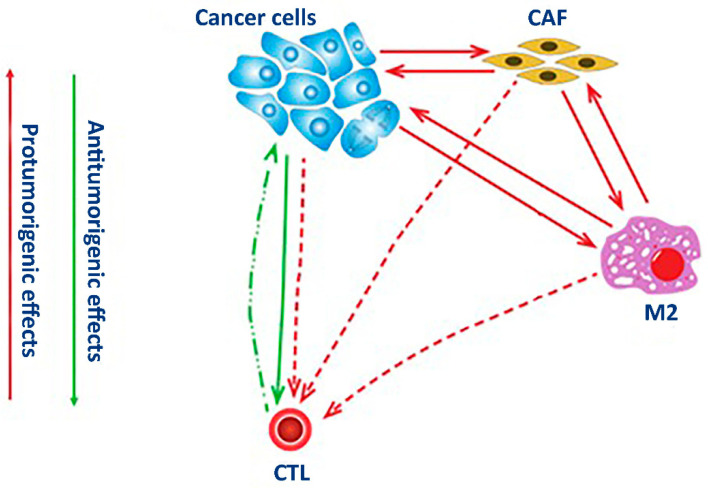
The logic model of NSCLC organoid tumor microenvironment. Cancer cells (CAN) promote the activation of stromal cells into cancer-associated fibroblasts (CAF) and the polarization of macrophages toward the M2 phenotype (M2). These cells stimulate each other and suppress the anti-tumor activity of cytotoxic lymphocytes—CTL (red arrows). Contrarily, their anti-tumor activity is stimulated by specific antigens on the surface of some tumor cells (green arrows). Solid lines represent cell stimulation, dashed lines—cell suppression, dash-dotted lines dots—cell destruction.

**Figure 2 life-13-02228-f002:**
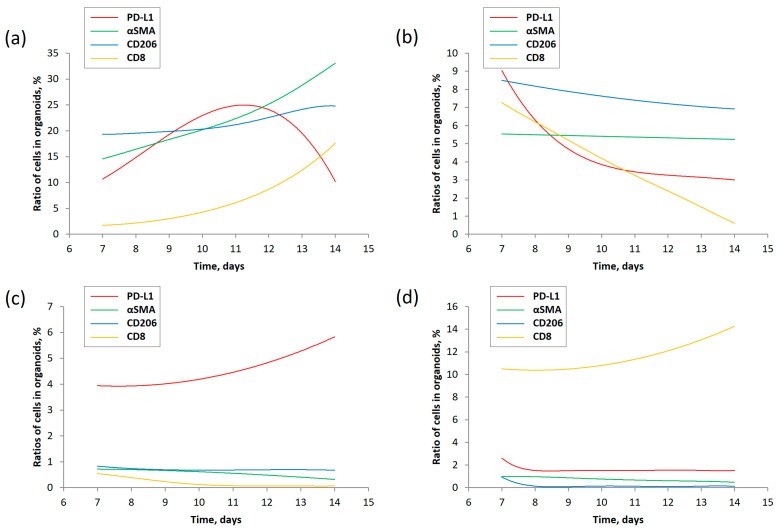
Model-predicted dynamics of cell quantity in organoid from 7th to 14th day of incubation. (**a**) Patient 8; (**b**) Patient 9; (**c**) Patient 13; (**d**) Patient 14. PD-L1—cancer cells, CAF—cancer-associated fibroblasts, M2—M2 polarized macrophages, CD8—cytotoxic T-cells.

**Table 1 life-13-02228-t001:** Cell-specific biomarkers used in flow cytometry experiments.

Type of Cells	Cell-Specific Biomarker
Cancer cells	PD-L1
Cancer-associated fibroblasts	αSMA
M2-polarized macrophages	CD206
Cytotoxic lymphocytes	CD8

**Table 2 life-13-02228-t002:** Previously assessed parameters of mathematical model.

Parameter	Definition	Published Value	References	Adjusted Value
*γ*	Growth rate of cancer cells	0.05–0.44 day^−1^	[17,18]	0.05 day^−1^
*K*	Final number of cancer cells	10^9^–3.3 × 10^9^ day^−1^	[19]	10^6^ day^−1^
*q* _1_	Stimulation of cancer cells by M2-polarized macrophages	0.4 day^−1^	[19]	4 × 10^−5^ day^−1^
*q* _3_	Stimulation of M2 macrophages by cancer cells	4 × 10^−8^ day^−1^	[19]	4 × 10^−8^ day^−1^
*δ_M_* _2_	Death rate of M2-polarized macrophages from natural causes	0.2 day^−1^	[18]	0.2 day^−1^
*k*	Number of cancer cells eliminated by cytotoxic cells	3.4 × 10^−10^–1 × 10^−3^ cell^−1^ day^−1^	[18]	0.001 cell^−1^ day^−1^
*δ_Tc_*	Death rate of cytotoxic cells	2 × 10^−3^–1 day^−1^	[18]	0.1 day^−1^

**Table 3 life-13-02228-t003:** Experimentally assessed parameters of mathematical model.

Parameter	Description	Calculated Values, Day^−1^
*q* _2_	Stimulation of cancer cells by cancer-associated fibroblasts	0.0001–0.005
*q* _4_	Stimulation of M2-polarized macrophages by cancer-associated fibroblasts	0.0001–0.001
*q* _5_	Stimulation of cancer-associated fibroblasts by cancer cells	0–0.00001
*q* _6_	Stimulation of cancer-associated fibroblasts by M2-polarized macrophages	0.00001–0.001
*q* _7_	Stimulation of cytotoxic T cells by cancer cells	0.0009–0.0015
*q* _8_	Suppression of cytotoxic T cells by M2-polarized macrophages	0–0.00001
*q* _9_	Suppression of cytotoxic T cells by tumor-associated macrophages	0–0.00001
*δ_CAF_*	Death rate of cancer-associated fibroblasts	0.1

## Data Availability

The data presented in this study are available on request from the corresponding author.

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
