# Peer review of "Mathematical Modeling of Non-Small-Cell Lung Cancer Biology through the Experimental Data on Cell Composition and Growth of Patient-Derived Organoids"

_life, 2023, doi:10.3390/life13112228_

Round 1
Reviewer 1 Report
Comments and Suggestions for Authors
This study proposed a mathematical model with the data on specific biomarkers of cancer cells (PD-L1), tumor-associated macrophages (CD206), natural killer cells (CD8), and fibroblasts (αSMA) as input, which can accurately predict the cellular composition of patient-derived tumor organoids at a desired time point. Combining the obtained results with “omics” approaches could improve the current understanding of the nature of non-small cell lung cancer (NSCLC). Moreover, their implementation into clinical practice could facilitate a decision-making process on treatment strategy and develop a new personalized approach in anti-cancer therapy. The topic fits the scope of the journal, and the references are updated. In general, this manuscript is fairly organized, and the experiments can support the conclusions in this manuscript. Key issues are required to be addressed before its publication on Life.
Major points:
1. In the introduction section, the rationale of this study is required to be clearly described.
2. Lots of introductory contents about the current development of mathematical models were put in the discussion section. The authors are required to re-organize the introduction section and discussion section. In the discussion section, this study-related contents can be discussed or compared with the previous reports, but the introductory contents are required to be put in the introduction section.
3. The current development status of mathematical models for the other cancer types are required to be briefly introduced in the introduction section.
Minor points:
1. The full name of NSCLC is required in Abstract when it shows up for the first time.
Comments on the Quality of English LanguageThe English language is fine.
Author Response
The revised manuscript with highlighted changes is attached with this reply (.docx file).
Reply
Thank you for allowing us to submit a revised draft of my manuscript entitled "Mathematical modeling of non-small cell lung cancer biology through the experimental data on cell composition and growth of patient-derived organoids" for publication in MDPI Life.
We appreciate the time and effort that you and the reviewers dedicated to providing feedback on our manuscript and are grateful for the insightful comments on and valuable improvements to our paper. We have incorporated most of the suggestions made by the reviewers. Please see below for a point-by-point response to the comments and concerns. All page numbers refer to the revised manuscript file with tracked changes.
Major point 1.
- In the introduction section, the rationale of this study is required to be clearly described
We agree with this comment. We have, accordingly, modified the introduction to describe the rationale behind the study more clearly.
Page 2, lines 54-65.
Major points 2 & 3
- Lots of introductory contents about the current development of mathematical models were put in the discussion section. The authors are required to re-organize the introduction section and discussion section. In the discussion section, this study-related contents can be discussed or compared with the previous reports, but the introductory contents are required to be put in the introduction section.
We agree with this comment. We reorganized and updated the text to incorporate the review paragraphs on NSCLC and other cancer models into the Introduction section, leaving only study-related comparisons in the Discussion.
Page 2, lines 54-71.
- The current development status of mathematical models for the other cancer types are required to be briefly introduced in the introduction section.
We agree with this comment and added an extra text to the Introduction.
Page 3, lines 54-65.
Minor point 1
- The full name of NSCLC is required in Abstract when it shows up for the first time.
Thank you for pointing this out. We replaced the acronym with the full name to reveal its meaning.
Page 1, lines 26-27.
We look forward to hearing from you shortly regarding our submission and to respond to any further questions and comments you may have.
Sincerely, Yegor Rumyantsev, the corresponding author.
Reviewer 2 Report
Comments and Suggestions for Authors
In this study, Sulimanov et al. proposed a mathematical model of non-small cell lung cancer biology through the experimental data on cell composition and growth of patient-derived organoids," the topic is both interesting and of significant importance with the enhancement of OMICs data retrieving and analyzing. The following comments and suggestions for enhancement are offered:
It would be beneficial to provide a brief overview of the current mathematical models for non-small cell lung cancer at the beginning of the paper, highlighting their limitations and the gap your study aims to fill.
Figure 2 presents the model-predicted dynamics of cell quantity in the organoid. To estimate the reliability and validity of the model, it's imperative to juxtapose these predictions with real experimental data. Comparing model predictions with actual observations will not only substantiate the model's accuracy but also highlight areas of improvement. In subsequent iterations or publications, consider including graphs or charts that display both predicted and observed values side by side. This would provide readers with a clear visual representation of the model's performance and its potential clinical implications.
Author Response
The revised manuscript with highlighted changes is attached with this reply (.docx file).
Reply
Thank you for allowing us to submit a revised draft of my manuscript entitled "Mathematical modeling of non-small cell lung cancer biology through the experimental data on cell composition and growth of patient-derived organoids" for publication in MDPI Life.
We appreciate the time and effort that you and the reviewers dedicated to providing feedback on our manuscript and are grateful for the insightful comments on and valuable improvements to our paper. We have incorporated most of the suggestions made by the reviewers. Please see below for a point-by-point response to the comments and concerns. All page numbers refer to the revised manuscript file with tracked changes.
Comment 1
It would be beneficial to provide a brief overview of the current mathematical models for non-small cell lung cancer at the beginning of the paper, highlighting their limitations and the gap your study aims to fill.
We agree with this comment and have updated the Introduction section with a brief overview of currently known NSCLC mathematical models. We have mentioned their limitations and highlighted our objectives.
Page 2, lines 52-79.
Comment 2
Figure 2 presents the model-predicted dynamics of cell quantity in the organoid. To estimate the reliability and validity of the model, it's imperative to juxtapose these predictions with real experimental data. Comparing model predictions with actual observations will not only substantiate the model's accuracy but also highlight areas of improvement. In subsequent iterations or publications, consider including graphs or charts that display both predicted and observed values side by side. This would provide readers with a clear visual representation of the model's performance and its potential clinical implications.
We agree with this comment and will be sure to perform validation of model’s predictive ability on additional experimental material and publish direct model-to-experiment comparisons in following works.
We look forward to hearing from you shortly regarding our submission and to respond to any further questions and comments you may have.
Sincerely, Yegor Rumyantsev, corresponding author.
Round 2
Reviewer 1 Report
Comments and Suggestions for Authors
Accept.
Comments on the Quality of English LanguageEnglish language is OK.
Reviewer 2 Report
Comments and Suggestions for Authors
Having reviewed the authors' responses and the amendments made to the manuscript, I am pleased to inform you that I find the current version of the paper to be satisfactory. The authors have adequately addressed my previous concerns, significantly improving the manuscript.
Specifically, the inclusion of a brief overview of currently known NSCLC mathematical models in the Introduction is commendable. This addition not only provides a comprehensive context for their work but also effectively outlines the limitations of existing models, thereby justifying the need for their study.
Furthermore, I appreciate the authors' commitment to performing validation of the model's predictive ability on additional experimental material and their intention to publish direct model-to-experiment comparisons in future works. This approach demonstrates a thorough understanding of the importance of validation in model-based research and adds credibility to their current findings.